# Development of Hydroxyapatite Coatings for Orthopaedic Implants from Colloidal Solutions: Part 2—Detailed Characterisation of the Coatings and Their Growth Mechanism

**DOI:** 10.3390/nano13182606

**Published:** 2023-09-21

**Authors:** Bríd Murphy, Mick A. Morris, Jhonattan Baez

**Affiliations:** 1Advanced Materials & Bioengineering Research Centre (AMBER), Trinity College Dublin, D02 CP49 Dublin 2, Ireland; baezj@tcd.ie; 2School of Chemistry, Trinity College Dublin, D02 PN40 Dublin 2, Ireland

**Keywords:** hydroxyapatite, solution deposition, novel coating, deposition kinetics, functional coatings, structural characterizations

## Abstract

This study is the second part of a two-part study whereby supersaturated solutions of calcium and phosphate ions generate well-defined hydroxyapatite coatings for orthopaedic implants. An ‘ideal’ process solution is selected from Part 1, and the detailed characterisation of films produced from this solution is undertaken here in Part 2. Analysis is presented on the hydroxyapatite produced, in both powder form and as a film upon titanium substrates representative of orthopaedic implants. From thermal analysis data, it is shown that there is bound and interstitial water present in the hydroxyapatite. Nuclear magnetic resonance data allow for the distinction between an amorphous and a crystalline component of the material. As hydroxyapatite coatings are generated, their growth mechanism is tracked across repeated process runs. A clear understanding of the growth mechanism is achieved though crystallinity and electron imaging data. Transmission electron imaging data support the proposed crystal growth and deposition mechanism. All of the data conclude that this process has a clear propensity to grow the hydroxyapatite phase of octacalcium phosphate. The investigation of the hydroxyapatite coating and its growth mechanism establish that a stable and reproducible process window has been identified. Precise control is achieved, leading to the successful formation of the desired hydroxyapatite films.

## 1. Introduction

The coating of metal titanium alloy implants (or other metals such as chromium cobalt) is key to improving the bodily response to a metal. As mentioned in Part 1 [1] of this study, hydroxyapatite (HA) coatings on orthopaedic implants mitigate fibrous build-up and promote implant fixation [2,3]. In this, Part 2 of the study, the evolution of HA as it grows on a substrate and the resulting film is extensively characterised. How HA film emerges with different mechanical properties from any coating process depends on many parameters, such as temperature, pressure and process time [4,5,6]. Similarly, HA will grow differently based on its precursors, be they synthetic or biologically derived; for example, mammalian or shell-derived HA can have variations in particle size or phase composition [7,8]. HA film will have different crystallinity based on its substrate chemistry [9,10,11]. Some methods of HA coating also incur water content which requires dehydration [12,13]. The interaction and the outcome between all these process parameters and the eventual film formed vary greatly by coating process.

There are many industrial techniques for HA coating, such as plasma spraying, chemical vapour deposition, electrochemical deposition, sol-gel deposition or ion assisted deposition, that can be used to form HA coatings on a surface. Within any synthetic process different calcium phosphate (Ca-P) phases can be created: amorphous calcium phosphate (ACP), α/β-tricalcium phosphate (α/β-TCP), octacalcium phosphate (OCP) or pure HA [14]. These other Ca-P phases have reabsorption and dissolution rates in vivo that differ from HA, thus affecting coating performance [15,16,17,18,19]. HA coatings must be sufficiently porous to encourage protein and mineral deposition in order to form new bone at the site [20,21,22,23,24].

The most common industrial HA coating techniques, plasma spraying and electrochemical deposition, have poor porosity control, poor crystalline phase control, require expensive tools (high temperatures or currents) and have poor coating adhesion [25]. Plasma spraying produces sub-micron needles and plates within micron-sized lamellae which allows for a porous structure [26,27]. Plasma spraying induces the less desirable α/β-TCP and relies upon post-deposition heating to crystalize the HA, which can result in low porosity [28,29]. Electrochemical HA coating deposits may sometimes suffer from defects in the coatings taking the form of certain Ca-P phases or hydrogen bubbles [30,31]. Sol-gel and biomimetic methods are commonly used in hydroxyapatite coating research [32]. Sol-gel processes can include elastic, part-polymer and ceramic composite materials which are useful to study the repair of bone cartilage [33,34,35]. Biomimetic processes using simulated bodily fluids have been studied as a means to grow HA on a surface [36,37]. Sol-gel deposited HA coatings have the highest coating adhesion of all methods, whereas biomimetic HA coating is the closest in nature to the body’s endogenous bone growth pathway [32,38]. Sol-gel and biomimetic processes are hindered by time-consuming steps and, in the case of biomimetic processes, rely heavily on the use of simulated body fluid [39,40,41,42].

Poor porosity and thickness control, along with poor substrate adhesion, are some of the unfavourable attributes of existing HA coating methods [43]. The advantages of existing methods lie within their mechanical properties and their easy affinity for titanium surfaces [44].

As outlined in Part 1 of this paper, this method of depositing hydroxyapatite (HA) on orthopaedic implant-type substrates, using saturated solutions of calcium and phosphorous, has the benefits of sol-gel techniques and biomimetic techniques. Coating crystallisation occurs at low temperatures, unlike in other methods. This process entices HA to grow in a self-assembling manner, encouraging high porosity and strict phase control. Part 1 of this paper provides us with an ideal process solution, process time and process temperature at which to grow hydroxyapatite coatings from solution. The formation of HA at surfaces is not well understood; the focus of research has mainly been on characterisation rather than the mechanism of film formation.

Further work is carried out herein on the mechanism of film formation upon a titanium alloy surface. Through thermal analysis, nuclear magnetic resonance (NMR), X-ray diffraction (XRD) and transmission electron microscopy (TEM), we identify the initial nucleation of Ca-P through to the final film characteristics. Once surface coating has commenced, XRD is performed at different stages of the process to identify each phase’s composition and how they emerge during the process. By proving the repeatable and reliable physiochemical outcomes of this process, a new generation of HA coatings can be engineered, ultimately improving patient prognoses post-implantation.

## 2. Materials and Methods

All materials and reagents were used as received. Monobasic potassium phosphate (KH_2_PO_4_) United States Pharmacopeia (USP) reference standard, Honeywell Fluka hydrochloric (HCl) acid solution 6 M, tris(hydroxymethyl)-aminomethane (TRIS) ACS reagent, 99.8% sodium chloride (NaCl) BioXtra and 99.5% calcium nitrate tetrahydrate (Ca(NO_3_)_2_·4H_2_O) ACS reagent were all from Sigma Aldrich (Wicklow, Ireland). A calibrated benchtop pH meter with a temperature enabled probe (Orion Star A111, [Thermo Scientific™, Loughborough, UK]) was used for pH measurements with an accuracy of 0.001 pH. Titanium coupons of Ti-6Al-4V alloy were used as substrates. All substrates were submerged in hot basic solutions to increase roughness and yield a negatively charged surface for the calcium ion attachment [45,46].

KH_2_PO_4_, TRIS and NaCl were mixed in deionized water (DIW) to yield a supersaturated phosphate concentrate. HCl was added to increase the stability of the concentrate to prevent precipitation. Ca(NO_3_)_2_·4H_2_O was mixed with DIW to yield a supersaturated calcium concentrate. For deposition, the supersaturated concentrates were combined before a dilution factor of between 15 and 17 was applied. The process solution was heated to 46 °C. A custom-designed experimental apparatus was used for deposition. A sample holder, thermometer and pH probe for in situ temperature and acidity measurements, and an overhead stirrer to agitate the solution, were inserted into the vessel. High agitation rates of 1000 RPM prevented gross precipitation of the apatite mineral from the solution. Substrates were placed in the reaction vessel for deposition then removed and rinsed with DIW; this process was repeated several times with fresh solutions to grow a coherent layer of HA at the solution–substrate interface. In between process runs, samples were left to dry for a minimum of 15 min in ambient conditions. To facilitate some powder analyses two powders were collected; (i) HA powder was collected by scraping it from surfaces post-solution deposition (for thermal analysis and nuclear magnetic resonance) and (ii) the process solutions underwent evaporation to remove the solvent and the dried solute was collected (for crystallinity analysis).

Solid-state nuclear magnetic resonance spectroscopy (NMR) was performed using a 9.4T Bruker Avance III HD NMR spectrometer equipped with a 3.2 mm H-F/X CP-MAS probe with TopSpin software version 3.6.5 (Bruker, Coventry, UK). The spectra were recorded at a Larmor frequency of 400.13 and 161.97 MHz for ^1^H and ^31^P, respectively. Chemical shifts were externally referenced to NH_4_(H_2_PO_4_) and TMS for ^31^P and ^1^H, respectively. A powder HA sample was added to a zirconia rotor, which was then spun at a magic angle spinning of 20 kHz. All spectra were run at room temperature. The ^31^P MAS spectra were measured at a spin rate of 20 kHz and with 50 kHz proton decoupling. An exponential window function of 20 Hz line broadening was applied to each FID before the Fourier transformation. The 2D ^31^P{^1^H} heteronuclear correlation (HETCOR) experiments were run with a contact time (τ_CP_) of 0.5 ms and a recycling delay of 8 s at 20 °C. Thermogravimetric analysis (TGA) was carried out using Perkin Elmer Pyris 1 TGA; samples were held at 30 °C for 15 min and then heated from 30 °C to 900 °C at 10.00 °C min^−1^. DSC was carried out using the Perkin Elmer Diamond DSC 800; the sample was heated from 20 °C to 500 °C at 10.00 °C min^−1^, held for 1.0 min at 500 °C before cooling to 20 °C at 100 °C min^−1^, held for 5.0 min at 20 °C and reheated to 500 °C at 10.00 °C min^−1^. However, it is the first heating which is reported on. X-ray Diffraction (XRD) patterns were acquired using a Bruker Advance Powder Diffractometer (Cu-Kα radiation with λ = 1.5406 Å, an operating voltage of 40 kV and a current of 40 mA). Measurements were performed in the 2θ range from 10° to 60° with steps of 0.004°. XRD diffractograms of HA powder from the dried process solution and HA films, after two, four and seven deposition runs, were collected. XRD was also performed on the blank titanium substrate. Scanning Electron Microscope (SEM) images were recorded using a Zeiss Ultra Plus system with the accelerating voltage of 5 kV, at a working distance between 3 and 10 mm and using an in-lens detector or secondary electron detector.

Lamellae for TEM cross-section images were prepared on a Zeiss AURIGA Focused Ion Beam (FIB), with accelerating voltages of 5–30 kV and ion beam currents of 50 pA–2 nA. Transmission Electron Microscopy (TEM) coupled with EDX analyses were performed on a FEI Titan 80–300 microscope with a Schottky-type field emission electron gun operated at 300 kV and a Bruker XFlash 6T-30 detector (resolution 129 eV).

## 3. Results

### 3.1. Analysis of HA Generated from within This Process in Powder Form

Powder analysis was performed on apatite mineral to understand how phases emerge within solution and upon solution–surface interaction. TGA analysis of this HA powder revealed the presence of adsorbed water and interstitial water, similar to other HA studies [47,48,49]. In the TGA data, Figure 1a showed a 4% weight loss by 600 °C at a steady rate. From 0 °C to 200 °C the weakly adsorbed water loss was ~2%. This was accompanied by a large heat flow in the DSC data (Figure 1b). Figure 1b showed a large peak at 113 °C which corresponded to a change in enthalpy of 101.2 Jg^−1^. The TGA data showed a decrease in mass from 200 °C to 400 °C likely due to chemically bound water loss of ~2%. Again, this was accompanied by a significant heat flow, shown in the DSC data by a smaller peak at 368 °C, which corresponded to a change in enthalpy of 29.7 Jg^−1^. There was no sign of phosphate decomposition, which only occurs significantly at temperatures >600 °C. The TGA analysis of HA compounds commonly show a weight gain at 700 °C upwards; this is a bounce back in weight due to the recrystallization of the mineral and it is seen in Figure 1a [50,51].

Solid-state NMR of HA powder was performed and the 1D spectra of ^1^H and ^31^P were collected (Appendix A). The 1D spectra gave little information about the hydration state of the phosphorous species because of low resolution. However, 2D heteronuclear correlation (2D HETCOR) experiments which correlated ^1^H chemical shifts with ^31^P (or other X-nuclei) chemical shifts provided excellent ^1^H resolution in the indirect dimension (Figure 1c). Cross-polarisation dynamics are different for ACP versus crystalline HA, due to different proton pair correlations [52,53,54]. ACP is correlated by the proton in water with the P proton of a phosphate: ^1^**H**_2_O
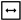
^31^**P**O_4_^3−^. Crystalline HA material is correlated by the proton of the hydroxyl functional group with the P proton of a phosphate O^1^**H**
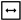
^31^**P**O_4_^3−^. Two distinct areas could be seen on the HETCOR spectrum (τ_cp_ = 0.5 ms) through contour vectors at the location of the cross peaks (Figure 1c). The cross-correlated peak at ^1^H 0 ppm was due to apatitic OH^−^ and can be assigned to a more well-formed crystalline HA. The broader cross peak between ^1^H 5 and 10 ppm was due to correlations between hydrogen phosphates, ACP and free and bound water, i.e., less well-formed HA. The NMR data indicated the existence of two distinct types of HA within the powder, one crystalline and one amorphous. The ACP peak also manifested the presence of water nuclei in and around all HA phases.

### 3.2. The Growth Mechanism of HA Film on a Titanium Substrate

Films were also analysed in situ to identify relationships between the properties of the solution and the films formed, plus characterisation of the film itself.

To identify the specific crystalline phases of HA present, extensive XRD analysis was performed (Figure 2a and tabulated in Table 1).

The titanium alloy has a strong reflection at a 2θ angle of 35.8°, denoted in Figure 2a as ‘Ti’. This titanium peak can be seen on HA coated substrates and the peak decreases in intensity as the number of HA deposition cycles increase [55,56]. There are three peaks of interest in this study with respect to HA [57,58,59,60,61,62]:OCP [002] is indicative of a calcium deficient plane [002] (26.2°);HA [210] (28.5°);HA Triple Peak containing the planes [211], [112] and [310] in varying degrees and could be masking the present of TCP which would be observed at a similar position.

The powder solute obtained directly from the process solution rather than the substrate gave an XRD pattern which shows amorphous build up around the peaks with a strong indication of triplet for HA. After two process runs, HA film on the substrate shows a weak OCP [002] peak and a strong characteristic HA triplet. This may suggest that the initial attachment could be TCP. After four process runs, the HA layer on the Ti substrate shows a strong OCP [210] peak, the emergence of a HA [210] peak and a broadening of the triplet peak with shoulders implying a mixture of planes. Clearly while HA is forming, the unstable phase of OCP begins to dominate within this solution process. Pure HA can have a peak around 26° but the intensity of it is lower than HA triple peak, unlike in this study where the OCP peak at 26° surpasses the HA triple peak intensity. By the 7th run of HA on the Ti substrate, the OCP [002] peak remains narrow and sharp and rises in intensity relative to the pure HA phases in the diffractogram. This implies that the coating grows preferentially along the [002] calcium-deficient plane within this process [63]. The comparison of diffractograms between HA film on the surfaces and dried HA solute shows that the substrate encourages much sharper peak shapes. This confirms that the presence of a substrate in solution causes heterogenous nucleation and orientation of the crystallites, with OCP being more prevalent than pure HA. This process could be driven by surface alignment or epitaxy between the substrate and the HA type coating.

The SEM images in Figure 2b,c represent the HA film at seven and two process runs, respectively. It is clear that the film develops cobweb-like interconnectivity with the pores, which develop to a full coverage of a porous film after seven process runs. This morphological structure has been shown in various in vitro studies to be the most advantageous HA structure by which to proliferate bone-growth cells [64,65,66]. As the coverage grows from Figure 2c to Figure 2b, it rationalises the diminishing Ti substrate peak.

It should be noted that OCP, Ca_8_(HPO_4_)_2_(PO_4_)_4_·5H_2_O, has an apatitic structure with a ‘hydrated’ water layer along the c axis [67,68]. These repeating OCP crystallites can be assigned to the [002] peak at XRD for our HA films. Correspondingly, our HA films also show the water content in the TGA, DSA and NMR data.

To further analyse the specific planes within the HA film on the titanium substrate, FIB lamellae were cut from the deposited films for cross-sectional TEM analysis. The TEM image indicates a coating thickness of 6 µm (see Figure 3a). Figure 3b–d shows <20 nm segments of dark and light features, which are thought to be layers of pores developed through the coating process. The dark areas are small nanoparticles, possibly gallium implanted during the FIB sectioning. Figure 3e,f shows crystalline areas circled, suggesting that the coating consists of polycrystalline domains within an amorphous matrix. The HRTEM images (Figure 3g,h) depict areas of high crystallinity with lattice fringes that are highly uniform and perfectly orientated with respect to each other. FFT analysis was performed on crystalline areas (Figure 3i). From the FFT diffractogram, we found the d-spacings of d = 0.344 nm and d = 0.284 nm correspond to the XRD peaks at 26° and 32° and correlate with the XRD data shown in Figure 2a [69,70]. In Figure 3i, strong [002] and [00$\bar{2}$] reflections are consistent with the XRD data. The absence of the [001] reflection suggests that here is still some of the pure HA phase present, since this phase has a symmetry assignment of P6_3_/m where this reflection phase is forbidden [71]. FFT diffractograms were fitted with a CIFfile 1534327, which is a mixture of OCP and HA, using CrystalMaker and CrystalDiffract software and showed excellent alignment (see Appendix A).

The crystallinity analysis presented here shows a coating comprising amorphous and distinct crystalline zones. The combination of these phases of ACP, OCP and some pure HA are desired for HA bone cement since these phases stimulate bone reformation in vitro [72,73]. New studies relating to the Mg alloy coatings for orthopaedic implants are also highlighting these ACP, OCP and HA phases as being beneficial for controlled resorption [74,75]. The phases, combined with the morphological structure of our HA coating, confirm that we have designed a process of making highly osteoinductive coatings.

## 4. Conclusions

This work has proved the efficacy of a coating process whereby an orthopaedic-quality film of hydroxyapatite can be generated at a surface from supersaturated solutions of calcium and phosphate ions. Based on findings in Part 1 of this study, films were generated herein using process solutions of specific concentrations. The subsequent growth of HA was investigated through various characterisation methods. Thermal analysis highlighted a hydrated nature within the HA film grown, which is to be expected from an aqueous process. However, some of the water content can be assigned to hydrated interstitial water content from the OCP phase. The NMR data proved that there are two distinct types of material being formed within this process: amorphous and crystalline.

Crystallinity analysis reveals the specific planes of HA that are present post-deposition. The XRD data show that the present of a substrate alters the formation mechanism and HA on a surface favours growth on the calcium deficient plane of [002] after seven process runs. The SEM data show the comprehensive HA coverage achieved, without compromising on the porosity of the film. The TEM data show that, after seven process runs, the thickness of the coating is 6 µm. The TEM data support that the film is an amorphous layer with crystalline pockets. The FFT analysis of these crystalline pockets show again that both OCP and HA are present through reflection and d-spacing data.

From these results, solution deposition of HA is shown to produce excellent HA coatings on titanium parts. The deposition can be tracked as the coating grows and the phase composition and morphology of the HA generated is advantageous for a low-temperature and -cost process. This body of work highlights that there is a large opportunity here in industrialising methods which have thus far only been used in research.

## Figures and Tables

**Figure 1 nanomaterials-13-02606-f001:**
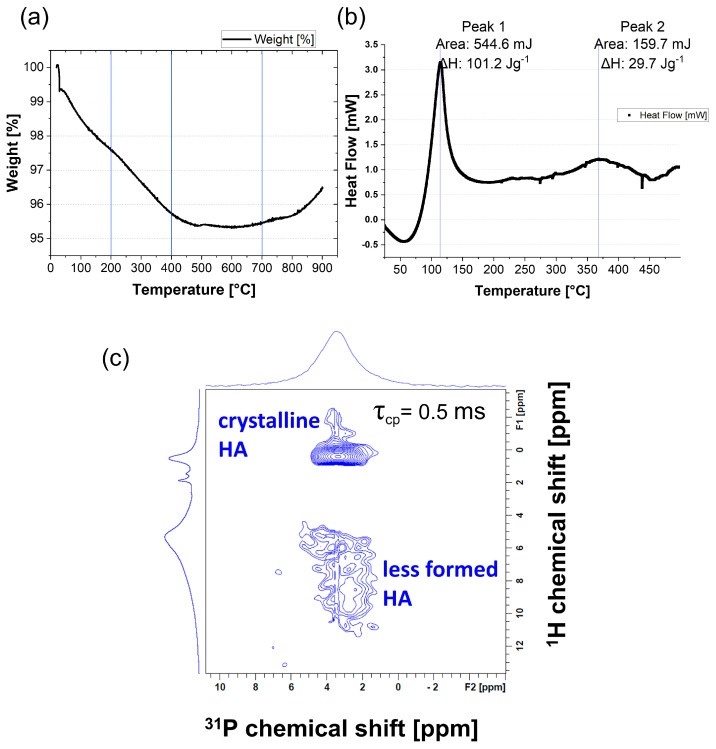
(**a**) Thermogravimetric data (TGA) showing weight loss as percentage loss from the samples versus temperature, (**b**) differential scanning calorimetry (DSC) data for the first heating of a sample, graph of heat flow (mW) versus temperature, (**c**) solid-state nuclear magnetic resonance 2D heteronuclear correlation sequence (NMR 2D HETCOR) measuring cross-polarisation of ^1^H and ^31^P proton spectra at a relaxation time (τ) of 0.5 ms, showing two distinct peak regions as blue contours.

**Figure 2 nanomaterials-13-02606-f002:**
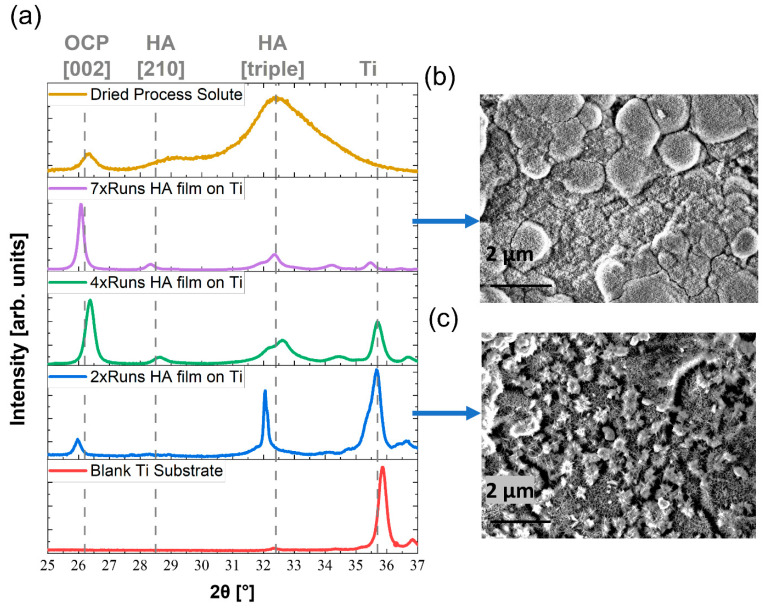
(**a**) X-ray diffraction (XRD) diffractograms in the region of 2θ angle 25° to 37°, with separate trends shown for a blank titanium substrate, films of hydroxyapatite on a titanium substrate after 2, 4 and 7 process runs and a trend from the dried process solute. (**b**) Scanning Electron Microscope (SEM) image of hydroxyapatite film on titanium alloy coupon after 2 hydroxyapatite solution deposition runs (**c**) SEM image of hydroxyapatite film on titanium alloy coupon after 2 hydroxyapatite solution deposition runs.

**Figure 3 nanomaterials-13-02606-f003:**
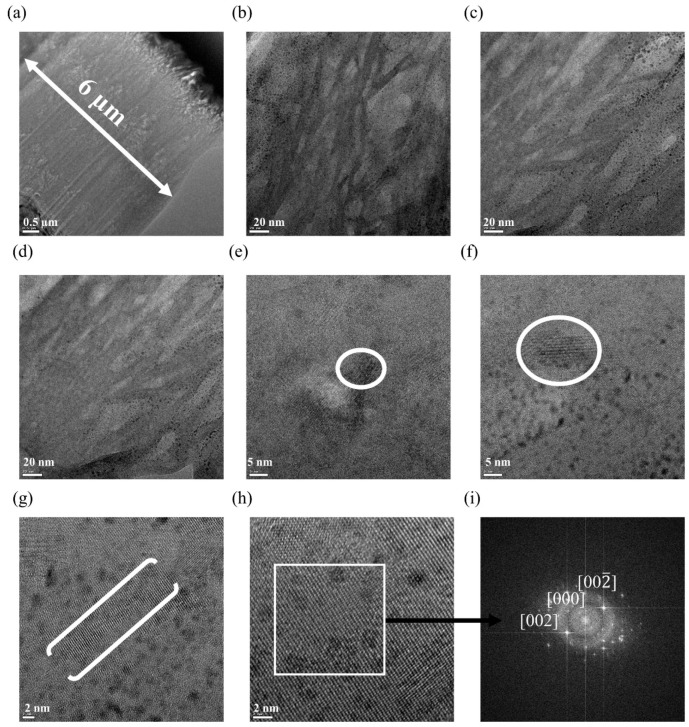
(**a**–**h**) Transmission electron microscope (TEM) images of lamellae cut into HA film over titanium substrate, (**e**–**g**) have areas of high crystallinity outline by white circles or brackets (**i**) fast Fourier transform (FFT) performed on crystalline areas which present in white square in (**h**) to identity lattice parameters.

**Table 1 nanomaterials-13-02606-t001:** Peak assignment of peaks seen in X-ray diffraction (XRD) diffractograms in Figure 2a.

	Peak Assignment			
Sample	OCP [002] (26.2°)	HA [210] (28.5°)	HA Triple Peak (32–33°)	Titanium (35.8°)
Dried process solute	Medium	Weak	Strong	Absent
7× runs HA film on Ti	Strong	Weak	Medium	Weak
4× runs HA film on Ti	Strong	Weak	Medium	Medium
2× runs HA film on Ti	Weak	Absent	Strong	Strong
Blank Ti substrate	Absent	Absent	Absent	Strong

## Data Availability

Data are contained within the article or the Appendix A.

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
