# Peer review of "Development of Hydroxyapatite Coatings for Orthopaedic Implants from Colloidal Solutions: Part 2—Detailed Characterisation of the Coatings and Their Growth Mechanism"

_nanomaterials, 2023, doi:10.3390/nano13182606_

Round 1

Reviewer 1 Report

The manuscript is Part II of a two-part study on hydroxyapatite coatings for orthopaedic im plants from colloidal solutions. It reports the results of detailed characterization of calcium phosphate coatings synthesized by sol-gel method using a novel procedure suggested in Part I published separately. Although I do not see much of a reason to split the study into two parts, the manuscript can be recommended for publication in Nanomaterials after addressing the following comments.

  1. A reference to Part I of this study should be added.
  2. Paragraph starting at Line 134. The goal of this paragraph is not clear. I suggest deleting it. If necessary some of this information can be added to the corresponding Sections below.
  3. Line 139. It is not a good idea to start this Section with the words “The aim of this section was to …” This sentence can be deleted or rephrased, for example, as follows: “To understand how phases emerge from a solution upon interaction with the surface, HA powder was studied…”
  4. Figure 1a. What is the origin of the weight gain at temperatures above 700 °C? Please, add its explanation to the text.   
  5. Figure 1c. It would be useful if the assignment of the two main peaks explained in the text were added to this image as well.
  6. Figure 2, caption. The description of the SEM method instrumentation should be moved to Section 2.
  7. Line 264. The phrase “… solution deposition of HA is shown to produce excellent HA coatings on titanium parts” needs to be substantiated by presenting earlier in the main text some data revealing why the coatings produced by this method are considered to be “excellent”. Some comparisons with the results obtained by other researchers using different techniques are desirable.
  8. Please, check the language. There are several mistypings in the text clearly originating from haste.

Please, check the language. There are several mistypings in the text clearly originating from haste.

Reviewer 2 Report

The authors should compare the differene of Part 1 and Part 2  and highlight the importance of the present work. Why did they report the works in one article? There are some similar statements such as introduction.

Reviewer 3 Report

The article "Development of hydroxyapatite coatings for orthopaedic implants from colloidal solutions, Part 2: detailed characterisation of the coatings and their growth mechanism." concerns the study of hydroxyapatite materials that can be used as well-defined hydroxyapatite coatings for orthopaedic implants. The title of the article corresponds to its content, and the subject matter is consistent with the profile of the journal.

The authors indicate the purpose of the research and its results. The references are well-chosen but works from recent years account for only 11% of all citations.

The presented results confirm the theses, they are correctly described. The TGA data shown in Fig 1A is not very clear. A typical TGA thermogram would be better. There are also no exact Figures captions contained in Suppl.

I believe that the article should be published after minor corrections:

1. supplementing References with works from recent years

2. change of Figure 1A to a clearer one

3. addition of Figures captions in Suppl.

Round 2

Reviewer 2 Report

The authors did not revised the statements that are similar with those in Part 1.

Round 3

Reviewer 2 Report

Accept